# Condenser Design for On-Board ORC Recovery System

Roberto Capata [1] and Giuseppe Piras [2,*]

1   Department of Mechanical & Aerospace Engineering, Sapienza University of Rome, 00184 Rome, Italy; roberto.capata@uniroma1.it
2   Department of Astronautics, Electrical and Energy Engineering, University of Rome, 00184 Rome, Italy
*   Correspondence: giuseppe.piras@uniroma1.it; Tel.: +39-06-445-85

**Abstract:** In this study, an ORC plant is considered that recovers the heat source from 1800 cc Diesel Engine exhaust gases. The power recovered by the ORC system could be used, as an auxiliary system, to provide energy to the electronic parts of the car. The purpose of this study is to investigate the possibility of allocating this ORC plant into a vehicle and in the design of the condenser, with the purpose of reaching the best configurations that lead to low-pressure drop and compact dimensions. The Organic Rankine Cycle converts thermal energy from low-grade heat sources into electricity. The thermodynamic cycles to produce 5 kW were simulated using a custom software application. The basic cycle was chosen to guarantee the most compact configurations. The selected thermodynamic parameters are based on the need to cope with atmospheric conditions. The working fluid chosen is R245fa, due to its thermodynamic characteristics. Since the electrical part would be partially satisfied by this system, the entire power of the thermal engine would be dedicated to mechanical use. It could also be assumed that, as a consequence of the settings of this plant, a reduction in fuel consumption could be expected, which, although at a percentage that cannot be theoretically evaluated at the moment, is nevertheless predictable. In this first stage of the evaluation, the condenser design is presented and analyzed.

**Keywords:** Organic Rankine Cycle; thermodynamic feasibility; condenser design

## 1. Introduction

Since the 1970s, the importance of new means of providing energy has grown. In addition, the recent Paris Agreement highlighted the urgent need for energy efficiency initiatives, together with the potential reuse of waste energy [1]. The EU Sustainable Development Goals require major changes in current energy systems, aimed at increasing flexibility [2] and reducing the consumption of energy resources [3]. The Organic Rankine Cycle is widely studied thanks to its numerous and interesting applications. The ability to produce energy from a relatively low temperature heat source is the core of all applications [4]. The most common plants are biomass [5], geothermal, solar, desalination. The Organic Rankine Cycle can be used to recover energy from exhaust gases, such as electric trains and boats, improving fuel consumption and reducing their environmental impact [6]. The interesting, perhaps new, aspect is that in almost all cases, when researchers talk about ORC systems, they are talking about stationary configurations [5–10]. Apart from that, there are a lot of works on small ORC systems. Some authors studied how to optimize—at these small rates—the heat exchangers [11]; others evaluated their technical and economic feasibility [12] and characteristic operating points [13]. In others, the feasibility of particular organic fluids and the use of volumetric expanders [14,15] have been studied. In this case, the application is vehicular, trying not to excessively modify the car structure and not to vary (or to vary as little as possible) both the payload and the weight arrangements to ensure the car's stability. Another aspect is to use the exhaust gases of the ICE engine as an energy source. This source varies depending on the driving cycle. Thus, the study proposed here is focused on the process of optimizing the ORC System and its various components

at the various engine speeds, accepting variations in performance (fixing them within a certain range, in our case 5%). The advantages of ORC are various. The organic substance used is usually characterized by a low boiling point, a low latent heat of evaporation and high density. These properties are preferable to increase the mass flow rate into the expander. Then, the specific heat of evaporation of the organic fluid is considered lower than that of water; this is the main reason why the organic fluids are used in place of water for heat recovery from sources in the low to medium temperature range. Another important factor that has contributed to the diffusion of the ORC is the possibility of adapting the same system for different heat sources with minor changes—the components used, in fact, can be derived from those of air conditioning, which have already reached full technological maturity. The favorable performance of ORCs in energy recovery, if adopted by industrial facilities, could ease electricity demand, simultaneously decreasing fossil fuel consumption and increasing the overall energy efficiency.

The temperature and pressure of the exhaust gases, in this case 723 K and a pressure of 1 bar, set a limit to the use of organic fluid and the net power. Once the operating fluids have been chosen, the pump, turbine, and condenser characteristics (dimensions, input and output temperature, pressure ratio, etc.) are calculated and the "nearly-optimal" combination is sought.

## 2. Cycle Analysis

Firstly, the study of the selected working fluid, R245fa (Pentafluoropropane), is necessary for the project (see Figure 1).

R245fa: HFC-245fa is also known as pentafluoropropane and its chemical name is 1,1,1,3,3-Pentafluoropropane. It has no ozone depletion potential and is nearly non-toxic [16,17]. The p-h diagram is shown below.

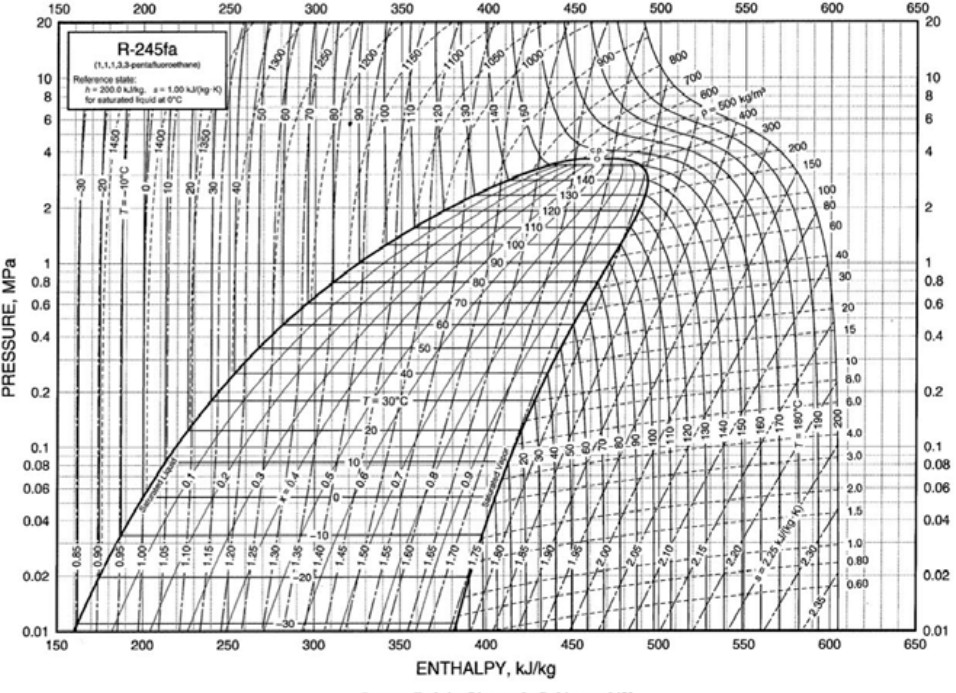

**Figure 1.** R245fa p-h diagram.

From the literature, it has been estimated that the temperature of the exhaust gas is 720 K (450 °C). This fluid was selected because it is easily available and all its physical and chemical characteristics, as well as its environmental impact, are well known. Other fluids have been studied on paper and some tested in the laboratory, such as R134 [18]. The choice was made on the basis of the available data and on the basis of very precise experimental

methodology and issues (availability, safety, etc.). Once the working fluid was defined, the following step was to create, with a customer plant cycle simulator, the basic ORC plant, with the following mandatory components: pump, turbine, heat exchanger, and condenser.

The process sequence (Figure 2) is numbered and described as follows:

1.   1–2: Expansion in the expander. The heat energy of the working fluid is converted into mechanical energy by an expander; then, an alternator (not represented) converts this mechanical energy into electricity.
2.   2–3: Condensation in a condenser. The vapour fluid condenses to the liquid state.
3.   3–4: Compression in a pump. A feed pump pressurizes the liquid working fluid.
4.   4–1: Vapourization in a boiler. The liquid working fluid absorbs thermal energy and vaporizes to the vapor state.

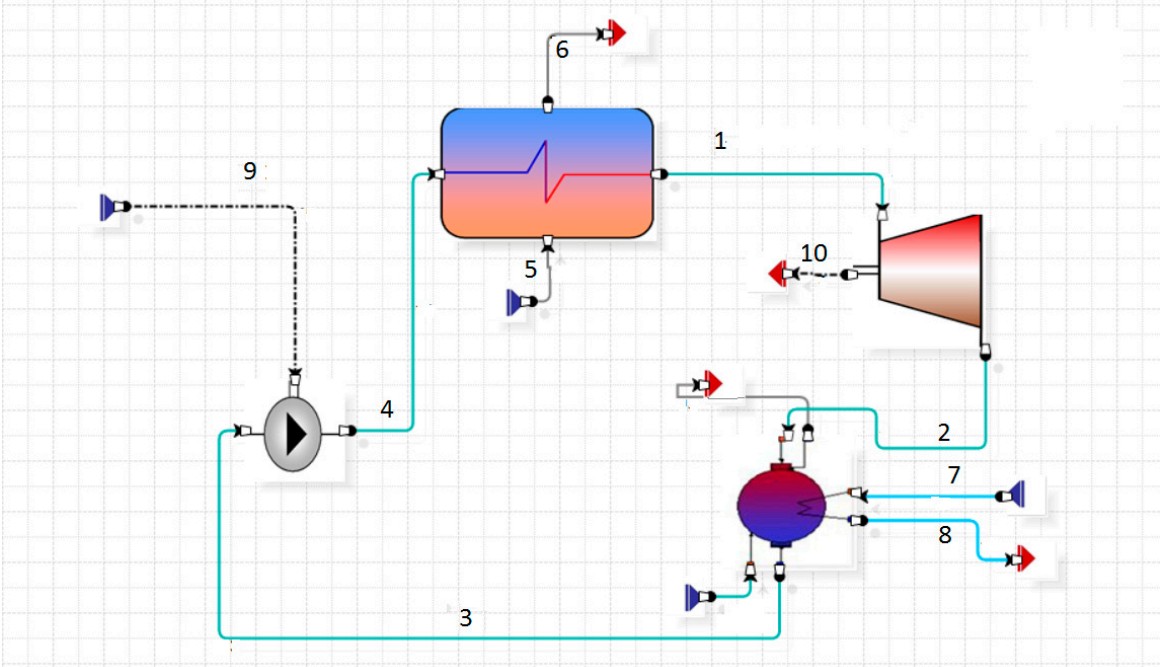

**Figure 2.** ORC plant on the simulator.

Flows number 1–4 describe the thermodynamic cycle, 5–6 are the exhaust gas inlet and outlet, 7–8 are the cooling water inlet and outlet, and 9–10 are the power input/output for the pump and the turbine, respectively. The simulation input values are listed below. The power is the input data and the others are the experimental values of the operating 1800 cc diesel engine (the engine data are supplied by a previous laboratory test in the University department and reported in several previously published papers).

1.   Power output: 5 kW.
2.   The exhaust gas inlet temperature (T = 723 K).
3.   The exhaust mass flow rate ($\dot{m}$ = 0.3 kg/s).
4.   The cooling water inlet temperature (T = 313 K).

The thermodynamic simulation is needed to identify the operational conditions of the components and to analyze—in this particular case—the operative parameters of the condenser. Thus, Table 1 reports the results of the simulation.

**Table 1.** ORC plant simulation results.

| R245fa Input Data | |
|---|---|
| mass flow rate (kg/s) | 0.5 |
| boiler inlet temperature (K) | 333 |
| boiler outlet temperature (K) | 390 |
| boiler inlet pressure (kPa) | 1250 |
| boiler outlet pressure (kPa) | 1225 |
| condenser inlet temperature (K) | 370 |
| condenser outlet temperature (K) | 341 |
| condenser inlet pressure (kPa) | 610 |
| inlet cooling water temperature (K) | 313 |
| outlet cooling water temperature (K) | 334 |
| power output (kW) | 5 |
| power absorbed by the pump (kW) | 0.233 |

*The Process Simulator*

The Process Simulator CAMEL-Pro™ is written in C++ and C#, is based on a completely—and authentic—object-oriented approach and is equipped with a user-friendly graphical interface that allows the simulation and analysis of several energy conversion processes. The system is represented as a network of components connected by material and energy streams; each component is characterized by a set of equations describing the thermodynamic changes imposed on the streams; in mathematical terms, this equation system is not closed, and, therefore, needs a proper number of boundary conditions in terms of known flow parameters. In practical terms, this means that the computed solution depends on both the plant configuration and on the assigned boundary conditions. An optimized Newton–Raphson iterative algorithm is used to solve the global equation system. The main feature of CAMEL-Pro™ is its modularity, which enables users to expand the code by adding new components or by modifying the model of the existing ones. In the gas model adopted in CAMEL-Pro™, the specific heat is calculated by a fifth-order polynomial in temperature, and enthalpy and entropy are obtained by the exact integration of these polynomials. The gas constant R is calculated according to the mixture rule. For water/steam properties, CAMEL-Pro™ uses the IAPWS library (The International Association for the Properties of Water and Steam, 1997). Other models for material streams are also available and the proprieties of R245 are derived from the Coolprop library.

**3. Thermodynamic Model of the Condenser**

The condenser is usually the power plant's largest component. Consequently, it is important to control its size and the pressure drop [19–22]. The chosen condenser has a particular configuration: the working fluid passes into the pipes and the cooling water embeds the shell. Cooling water is always in circulation and a recirculation pump is used. As an initial configuration, this pump circulates water from the radiator to the condenser.

The condenser dimensions strictly depend on the overall heat coefficient U (maximum value adopted). Besides, the U coefficient depends on the exchange area and the heat transfer coefficients. Optimizing U is difficult because a compromise between multiple parameters has to be found. Condensation occurs in three different phases:

1. The de-superheating, where the fluid's temperature almost reaches the condensation temperature.
2. The condensation phase, where the gaseous working fluid becomes liquid.
3. The sub-cooling phase is used to slightly decrease the temperature in order to make sure that all the fluid is liquid.

The condenser has been theoretically divided into these phases; each phase has been evaluated and designed on the basis of the simulator's results. In each part, a turbulent stream is considered in order to maximize heat exchange. The cooling fluid is water. To maximize the heat exchange and reduce the component dimensions, copper elements

were used. This material provides excellent thermal performance and is compatible with refrigerants. The aim, as mentioned above, is to minimize the overall condenser dimensions and maximize the pressure drop. The following figure shows the condenser as it appears in the plan configuration.

Considering Figure 3, the streams numbered 1 and 2 are the inlet and outlet of the working fluid, 3 is linked to a tank (that could fill the circuit when needed; it is not present in the plant shown), and 4 and 5 are the streams representing the inlet and outlet of the cooling fluid. In this study, we operated in the most critical condition by adopting a temperature of 40 °C (which would correspond to a typical summer situation). The geometry of the condenser will be square, so the figure is for reference only. The various possible methods for designing the condenser are briefly outlined and analyzed below.

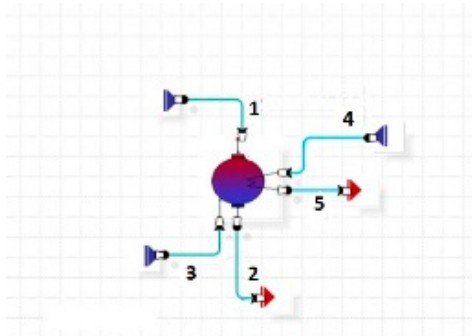

**Figure 3.** Detail of the condenser software representation taken from Figure 2.

*3.1. LMTD Model*

This method is used to define the condenser when the inlet and outlet temperatures are known. The LMTD (Logarithmic Mean Temperature Difference) is defined as:

$$LMTD = \Delta \text{Tlm} = \frac{\Delta T_1 - \Delta T_2}{ln \frac{\Delta T_1}{\Delta T_2}} \tag{1}$$

where $\Delta T_1$ is the difference between the inlet temperature of the hot source and the outlet temperature of the cold source and $\Delta T_2$ is the difference between the hot outlet temperature and the cold inlet temperature.

$$\Delta T_1 = T_{h,i} - T_{c,o} \tag{2}$$

$$\Delta T_2 = T_{h,i} - T_{c,o} \tag{3}$$

The condenser surface can be determined by the following equation:

$$A = \frac{Q}{U \ LMTD} \tag{4}$$

The formula shows that U and A are inversely proportional. That means that it is necessary to maximize U in order to obtain a minimum exchange surface. The overall heat transfer, given by combined conduction and convection, is frequently expressed in terms of an overall heat-transfer coefficient "U". For a cylinder, the geometry is given by:

$$U = \frac{1}{\left(\frac{1}{h_i}\right) + \frac{d_i \ln\left(\frac{d_o}{d_i}\right)}{2k} + \left(R_A \frac{1}{h_{water}}\right)} \tag{5}$$

The terms "$d_i$" and "$d_o$" represent the diameter of the inside and outside areas of the inner tube. To decrease the number of unknowns, the calculation of heat transfer coefficient "U" is only limited to the inner area "$A_i$", and the term "$A_i/A_o$" is replaced by the equation:

$$A_i = \pi d_i L_1$$

*3.2. The Heat Exchanger Method*

HEEM offers an alternative to optimize the design. In this case, various heat exchangers can be compared to select the optimal configuration corresponding to the optimum efficiency. Efficiency is defined as the ratio of the actual heat transfer to the maximum possible heat transfer. The following formula can be used to calculate the effective heat transfer in a reverse flow configuration:

$$\varepsilon = \frac{actual\ heat\ transfer}{maximum\ possible\ heat\ transfer} \tag{6}$$

The maximum value of the heat transfer is expressed by the maximum heat received by the fluid. It is defined by the minimum value of the product of mass and specific heat.

$$Q = \dot{m}\ cp\ \Delta T \tag{7}$$

In addition, the NTU (number of transfer units) can be used to calculate effectiveness in a different way.

$$NTU = \frac{U\ A}{(m\ cp)_{min}} \tag{8}$$

In this case, we have two different effectiveness values depending on the evaluated phase. For the monophasic scenario, the effectiveness is expressed by the formula:

$$\varepsilon = \frac{actual\ heat\ transfer}{maximum\ possible\ heat\ transfer} \tag{9}$$

In the condensation phase:

$$\varepsilon = 1 - e^{NTU} \tag{10}$$

## 4. Design Procedure

The chosen configuration (Figure 4) is the "circular tube fin". It has several advantages such as reduced weight, better temperature control and easier transport. The type of tube is often chosen to reduce losses [13–15]. The inner diameter of the tube is 8 mm, the fin length is 3 mm, the tube thickness is 1 mm and the distance between two consecutive fins is 2 mm. The staggered arrangement is triangular, to avoid interference problems, and the distance between each arrangement center is 17 mm (the distance will be called $p_t$ or $p_l$ if it is from tube to tube or from tube to the outer shell).

After the number of tubes is set, the width, thickness, and length are defined. At this point, we have to distinguish what happens in the mono-phase or the two-phase condensation. The analysis will be conducted for both the working fluid (inside) and the water (outside).

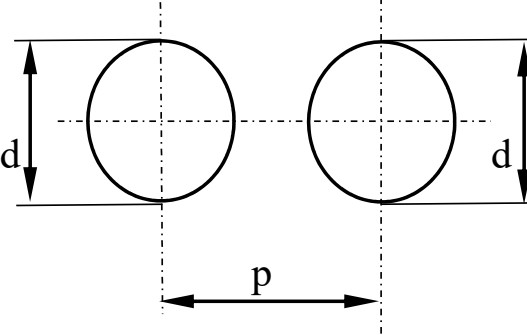

**Figure 4.** Reference scheme.

### 4.1. Monophasic Condensation

Working fluid side

The fluid characteristics (viscosity coefficient, thermal conductivity, etc.), were derived from the Coolprop library. First, the enthalpy difference is calculated to compute the thermal exchange. Then, the LMTD is derived with the method previously described. It is now necessary to compute the overall heat coefficient U in order to find the required exchange area. To compute the coefficient U we have to introduce the basic dimensionless numbers: Reynolds, Nusselt, Prandtl, and Froude. The **Reynolds number** (Re) is defined as:

$$Re = \frac{\rho u_{\mathrm{m}} D_{\mathrm{h}}}{\mu} \tag{11}$$

where $\rho$ is the fluid density, $u_{\mathrm{m}}$ is the fluid velocity, $\mu$ is the dynamic viscosity, and $D_{\mathrm{h}}$ is the hydraulic diameter (it will be different in the working fluid or cooling fluid case).

The **Nusselt number** (Nu) is a dimensionless number expressed by the ratio convective to conductive heat transfer. A Nusselt number close to one is characteristic of "slug flow" or laminar flow. A larger Nusselt number corresponds to more active convection, with turbulent flow typically in the 100–1000 range.

$$Nu = \frac{h D_{\mathrm{h}}}{\lambda} \tag{12}$$

The Dittus and Boelter equation, which can be applied to the inlet region (where turbulent flow is not developed), will be used to compute the Nusselt number:

$$Nu = 0.023 Re^{0.8} Pr^{1/3} \left(\frac{d_{\mathrm{i}}}{L_1}\right)^{0.0055} \tag{13}$$

This equation is only used for single-phase operating.

The **Prandtl number** (Pr) is a dimensionless number, defined as the ratio of momentum diffusivity to thermal diffusivity. That is, the Prandtl number is given by:

$$Pr = \frac{\mu c_{\mathrm{p}}}{\lambda} \tag{14}$$

where $\mu$ is the dynamic viscosity, $c_{\mathrm{p}}$ is the specific heat capacity and $\lambda$ is the thermal conductivity. For gases, the Prandtl number varies from 0.2 to 1; for water or liquid, it varies from 1 to 10.

The **Froude number** (Fr) is a dimensionless number defined as the ratio of the flow inertia over the external field forces (the latter in, many applications, is simply due to gravity). The Froude number is based on the speed–length ratio, and it is defined as:

$$Fr = \frac{u}{\sqrt{g \cdot l}} \tag{15}$$

where $u$ is a characteristic flow velocity, $g$ is the gravity and $l$ is the characteristic length.

Once all quantities have been defined, the following procedure will be used.

1. Firstly, the Prandtl number is computed, and secondly, the calculation of the fluid velocity inside the pipes is evaluated, permitting the evaluation of the Reynolds number, which leads to the Nusselt numbers with Equation (13).
2. After computing the Nusselt numbers, it is then possible to estimate the $h_{\mathrm{i}}$ that is the heat transfer coefficient by Equation (12).
3. At this point, it is necessary to introduce the areas. When the numbers of tubes are known, where $N_{\mathrm{s}}$ is the number of pipes where the mass flow condensate, $N_{\mathrm{r}}$ is

the numbers of transits of the same bundle of tubes, and the total number $N_t$, the geometrical properties can be calculated. The inner area of the pipes is:

$$A_i = \pi d_i \, L_1 \, N_t \tag{16}$$

and the minimum free flow area is:

$$A_{o,i} = \left(\frac{\pi}{4}\right) d_i N_s \tag{17}$$

Cooling fluid side

As previously mentioned, the characteristics of the fluids are derived from the Coolprop library. The main areas (external pipes side) are the $A_p$, the primary area, the fin area $A_f$ and the heat transfer surface area $A_o$. The fin area is defined as:

$$A_f = \left[\frac{2\pi\left(d_f^2 - d_r^2\right)}{4} - \pi d_f d_f\right] N_f L_1 N_t \tag{18}$$

The primary area is the difference between the pipe surface area and the area blocked by the fins.

$$A_p = \pi d_r (L_1 - L_1 N_f t_f) N_t + 2 L_2 L_3 - \left(\frac{\pi}{4} d_r^2 N_t\right) \tag{19}$$

The $A_o$ is the total heat transfer area, computed by the sum of the primary area and the fin area.

$$A_0 = \pi \, d_r (L_1 - L_1 \, N_f t_f) N_t + 2 L_2 L_3 - \left(\frac{\pi}{4} d_r^2 N_t\right) - \left[\frac{2\pi\left(d_f^2 - d_r^2\right)}{4} - \pi d_f t_f\right] N_f L_1 N_t \tag{20}$$

Another important parameter to determine is the minimum areas among the pipelines, where the water flows. If a triangular configuration is chosen [10], it is possible to consider that surface, as a flat surface. In this way, the following simplified formula can be used, without significant errors.

$$A_{min} = [(P_t - d_r)L_1 - (d_f - d_r)t_f N_f L_1]\left(\frac{L_3}{P_t}\right) \tag{21}$$

The minimum area is represented in Figure 5.

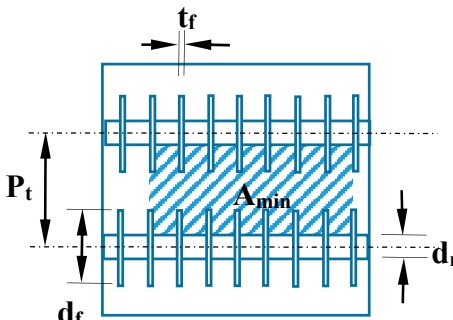

**Figure 5.** Minimum area description of the condenser.

The hydraulic diameter is calculated as follows:

$$D_0 = \frac{4 A_{min}}{\frac{A_p}{L_2}} \tag{22}$$

The Reynolds number for the cooling fluid is finally calculated with the hydraulic diameter. The velocity in the Re number is expressed by:

$$u = \frac{\frac{\dot{m}_{\text{water}}}{\rho_{\text{water}}}}{\frac{A_{\text{min}}}{N\text{r}}} \tag{23}$$

Finally, in this case, the Nusselt number changes and it is defined by the equation:

$$Nu = 0.134 Re_{\text{d}}^{0.681} Pr^{\frac{1}{3}} \left(\frac{s}{lf}\right)^{0.2} \left(\frac{s}{tf}\right)^{0.1134} \tag{24}$$

Collecting all obtained data, the overall heat transfer coefficient U can be computed:

$$U = \frac{1}{\left(\frac{1}{h_{\text{i}}}\right) + \frac{d_{\text{i}} \ln\left(\frac{d_{\text{o}}}{d_{\text{i}}}\right)}{2k} + \left(R_{\text{A}} \frac{1}{h_{\text{water}}}\right)} \tag{25}$$

Using the LMTD approach, the surface required for the heat exchange is defined by:

$$A = \frac{Q}{U \Delta Tlm} \tag{26}$$

where Q is the power exchanged, U the overall heat transfer coefficient, and $\Delta Tlm$ is the LMTD.

Pressure drop

In this paper, the pressure drop equation, proposed by Fanning has been used, with the Fanning friction factor "f" proposed by Taitel and Dukler:

$$\Delta p = \frac{4\rho f L_{\text{e}} u}{2d_{\text{i}}} \tag{27}$$

In Equation (28), $L_{\text{e}}$ is the equivalent length of the pipe, $\rho$ is the inner fluid density, u is the pipe fluid velocity and $d_{\text{i}}$ is the internal pipe diameter. The f factor is a function of the Re and the roughness of the pipe. For smooth pipes, the friction factor, in a range of $3000 < \text{Re} < 10^5$, can be approximated by:

$$f = \frac{0.079}{Re^{0.25}} \tag{28}$$

If $\text{Re} > 10^5$, the following equation is more accurate:

$$f = \frac{0.046}{Re^{0.2}} \tag{29}$$

### 4.2. Biphasic Condensation

As the condensation process proceeds along the pipes, the working fluid velocity decreases. At first, the condensation will occur on the wall of the pipes, then layer by layer, the liquid phase will increase. When the fluid is condensing, its thermodynamic characteristics change. The same procedure for the mono-phase has been adopted, introducing some necessary changes [22–24]. The evolution of the working fluid, from quality "0" to quality "1", has been divided into four-parts.

1.  Part one, when the quality **x** is within $0 \div 0.75$.
2.  Part two, when x = $0.75 \div 0.5$.
3.  Part three, when x = $0.5 \div 0.25$.
4.  Part four, when x = $0.25 \div 0$.

Similarly, for the cooling fluid side, there will be four corresponding stages. The four stages of the cooling water have been computed, assuming that every property is changing

linearly. From the working fluid side, the Martinelli parameter is introduced to compute the Nusselt number. The formulae used for the R245fa is:

$$Nu = 0.023 Re^{0.8} Pr^{0.3} g(Xtt) \tag{30}$$

Re is the equivalent Reynolds number defined as:

$$Re_{eq} = \frac{G_{eq} D_h}{\mu_f} \tag{31}$$

where $G_{eq}$ is the liquid equivalent mass flow rate, expressed by the following equation:

$$G_{eq} = G \left[ (1 - x) + \left( \frac{\rho_l}{\rho_g} \right)^{0,5} \right] \tag{32}$$

x is the quality, and the $\rho_l$ and $\rho_g$ are, respectively, the liquid and the gas density.
Pressure drop

The approach to predict the pressure drop is the "homogeneous model", based on the experimentally derived theory. The pressure drop equation used is based on Kedzierski and Goncalves' theory, modified by Pierre, for the refrigerant fluid approach:

$$\Delta p_{tp} = \Delta p_{friction} + \Delta p_{accelleration} = \left[ \frac{f_N L_e (v_{out} + v_{in})}{D_h} + (v_{out} - v_{in}) \right] G^2 \tag{33}$$

where $v$ is the specific volume of the two-phases fluid, $L_e$ is the equivalent length and $f_n$ is the new friction factor:

$$f_N = 0.00506 Re^{-0.0951} K_f^{0.1554} \tag{34}$$

This $f_n$ is based on the Reynolds and $K_f$ number, when the fluid is all liquid, so Re is defined as:

$$Re = \frac{G D_h}{\mu_f} \tag{35}$$

The $K_f$ number is expressed by:

$$K_f = \Delta x \frac{\alpha}{Lg} \tag{36}$$

where $\alpha$ is the latent heat, provided by the Fluid-pro database, $\Delta x$ is the quality of the gas and g is the gravity.

## 5. Design Results

The system geometry has been defined respecting the required constraints. A square configuration of the condenser has been chosen. The previous Figure 3 represents the simulated condenser. Streams numbers 1 and 2 are the inlet and outlet of the working fluid, and streams 4 and 5 are the inlet and outlet of the cooling fluid. Stream 3 is the refill of the fluid; in this case, it is not considered, but it is important to mention the fact that this opportunity exists. Table 2 reports the operating specifications of the main streams.

After numerous iterations, the optimal solution has been found. The considered configuration is a compromise between a reasonable pressure drop and the smallest area. All data and a representation of the condenser are, respectively, reported in Table 3 and Figure 6.

**Table 2.** Condenser Streams.

| Stream n° 1 | |
|---|---|
| Temperature (K) | 371 |
| Pressure (kPa) | 610 |
| **Stream n° 2** | |
| Temperature (K) | 342 |
| Pressure (kPa) | 606 |
| **Stream n° 3** | |
| Temperature (K) | 313 |
| Pressure (kPa) | 101.3 |
| **Stream n° 4** | |
| Temperature (K) | 334.3 |
| Pressure (kPa) | 100.3 |

**Table 3.** The condenser's main dimensions.

| | |
|---|---|
| $L_1$ (m) | 0.3 |
| $L_2$ (m) | 0.3 |
| $L_3$ (m) | 0.2 |
| $d_i$ (m) | 0.008 |
| $d_r$ (m) | 0.01 |
| $d_f$ (m) | 0.016 |
| $P_t$ (m) | 0.017 |
| $P_l$ (m) | 0.017 |
| S (m) | 0.002 |
| $t_f$ (m) | 0.0003 |
| $N_r$ | 17 |
| $N_s$ | 15 |
| $N_{tot}$ | $17 \times 15 = 255$ |

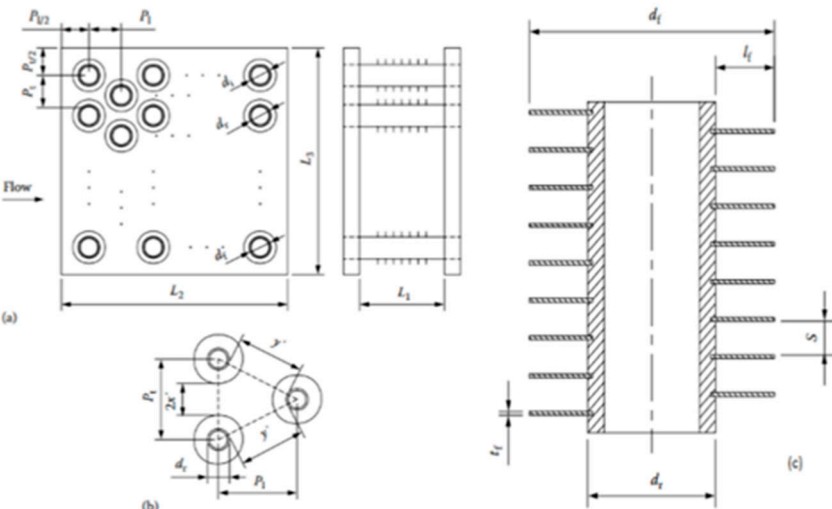

**Figure 6.** The condenser's chosen configuration. (**a**) actual disposition in the condenser assembling, (**b**) triangular configuration characteristics, (**c**) finned pipe representation.

Finally, Table 4 shows the operating conditions of the two fluids, R245fa and condensation water, in all its constituent parts (de-superheated, condensation, and sub-cooling).

**Table 4.** Fluid specifications.

| | R245fa | |
| --- | --- | --- |
| | **Inlet** | **Outlet** |
| **Mass flow rate (kg/s)** | 0.5 | 0.5 |
| **Temperature (K)** | 371.4 | 341 |
| **Pressure (kPa)** | 615 | 610 |
| **Enthalpy** | 60.3 | 32.5 |
| **Density** | 31.33 | 32.78 |
| **Steam quality** | 1 | 1 |
| | **Water** | |
| | **Inlet** | **Outlet** |
| **Mass flow rate (kg/s)** | 1 | 1 |
| **Temperature (K)** | 313 | 334.3 |
| **Pressure (kPa)** | 101 | 100.97 |
| **Enthalpy** | −2409.21 | −2403.55 |
| **Density** | 828.07 | 828.07 |
| **Steam quality** | 1 | 1 |

The simulation program also makes it possible to graphically show (Figure 7) the heat exchanged in each considered section. Once these quantities are known, it is possible to obtain the actual heat exchange surface in the various parts (Table 5) using the above procedure.

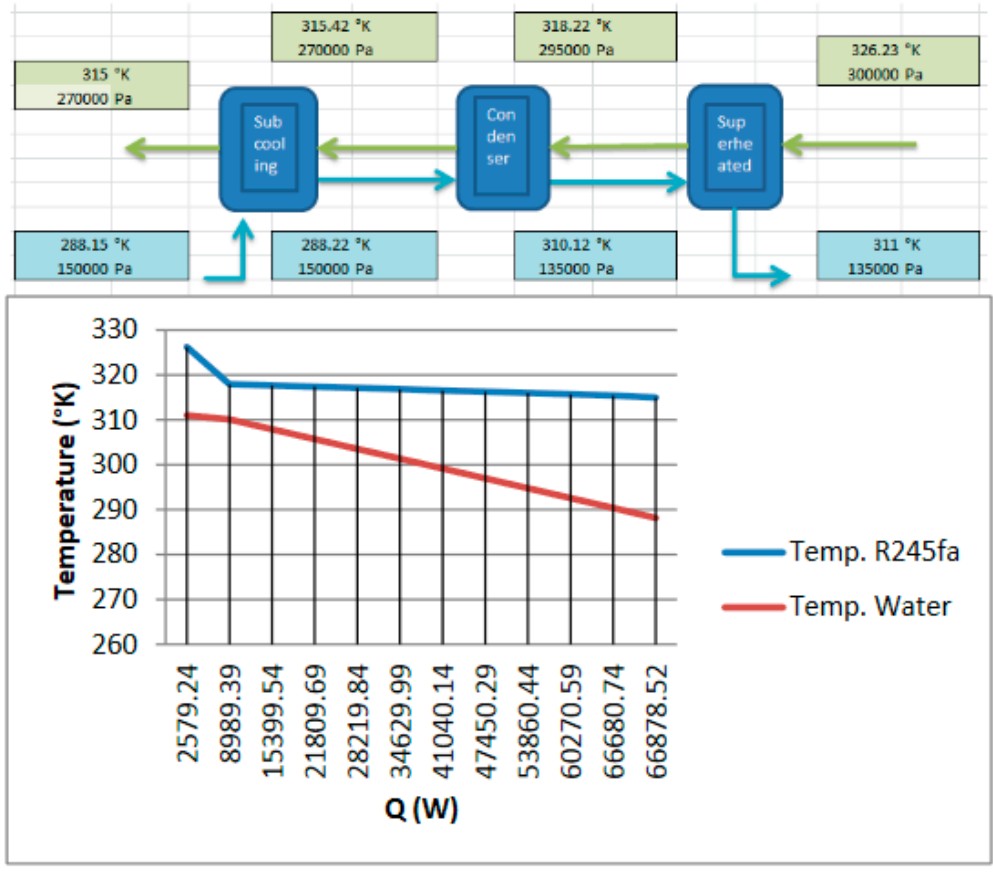

**Figure 7.** Thermal configuration and thermal exchange in the condenser.

**Table 5.** Heat exchanged and exchange surface.

| Heat Exchanged | | Heat Exchanger Surface | |
|---|---|---|---|
| de-superheated (kW) | 13.9 | de-superheated (m$^2$) | 0.27 |
| condensing (kW) | 80.9 | condensing (m$^2$) | 1.51 |
| subcooling (kW) | 0.8 | subcooling (m$^2$) | 0.02 |
| total (kW) | 95.6 | total (m$^2$) | 1.8 |

Finally, in order to evaluate the average effectiveness value, it is possible to weigh the effectiveness with the corresponding area. The procedure is the same as the one followed for the overall heat transfer coefficient.

$$\varepsilon = \frac{\Sigma(\varepsilon A)}{Atot} = 0.74 \tag{37}$$

$$U = \frac{\Sigma(UA)}{Atot} = 1734.5 \frac{W}{m^2 K} \tag{38}$$

The total pressure drop is the sum of each stage pressure drop:

$$\Delta p_{\text{tot}} = \Delta p_{\text{de-superheater}} + \Delta p_{\text{condensing}} + \Delta p_{\text{subcooling}} = 51,401.4 \ Pa \tag{39}$$

The inner area is:

$$A_{\text{i}} = \pi d_{\text{i}} L_1 N_{\text{t}} = 1.92 \ m^2 \tag{40}$$

The total area is slightly larger than the computed area. Thus, the component design can be accepted and valid. Figure 8 represents a 3D view of the condenser with inlet/outlet streams.

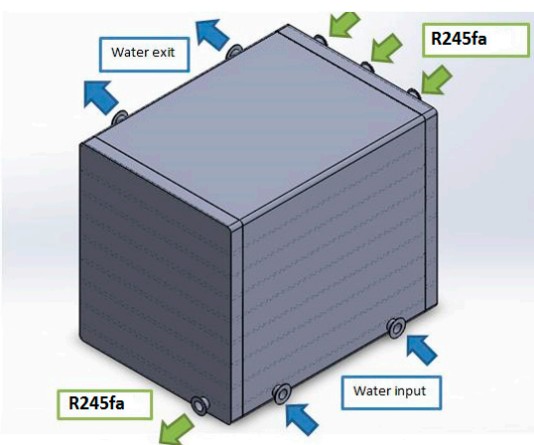

**Figure 8.** Condenser 3D assembled view.

## 6. The On-Board Configuration

An empirical method has been used to demonstrate that the proposed ORC system can be mounted on a vehicle. In a commercial vehicle powered by methane, respecting the available spaces and volumes, the tank has been replaced and modified. For the positioning in the vehicle, since the HRSG has not been measured (at present), it has been assumed to be the same size as the condenser. It is known that the HRSG device is usually smaller than the capacitor. Figures 9 and 10 represent the possible plant configuration in a commercial vehicle. It is important to remember that it is necessary to realize an auxiliary circuit for the water cooling.

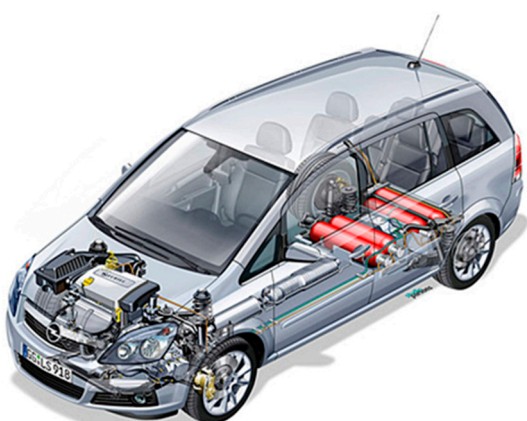

**Figure 9.** Commercial methane-fuelled vehicle (tanks in red).

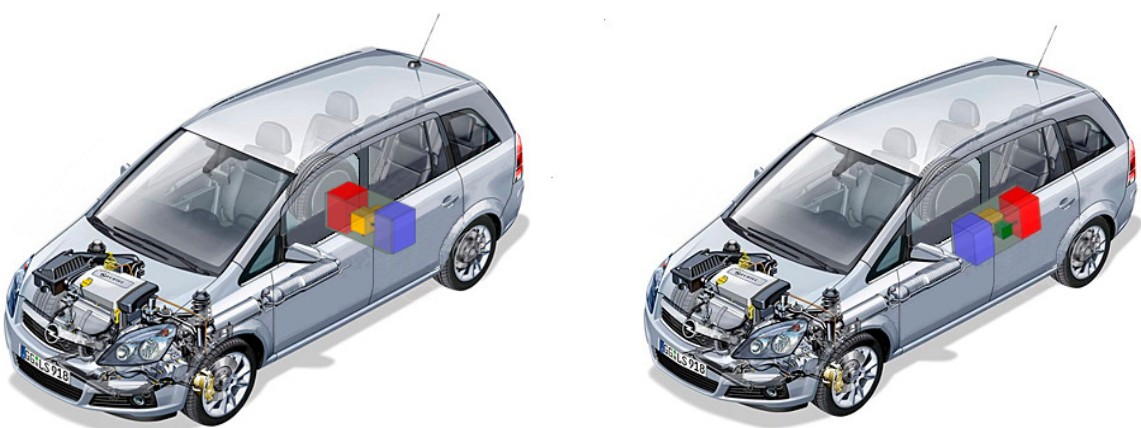

**Figure 10.** Possible allocation of the ORC plant inside the vehicle.

The additional radiator is of an air cooler type. This device is needed to extract heat from the ORC system and is smaller than the ones generally used on ICE vehicles. This component has a flow rate of about 6 L and can be found on the market; it is used in motorcycles with typical dimensions of $25 \times 25 \times 10$ cm. It is also important to remember that a detailed study of the additional radiator must be carried out without compromising the performance of the main radiator. The idea is to put the radiator in front of the car since it needs air to cool the water. Figure 11 shows the diagram of the system with the radiator.

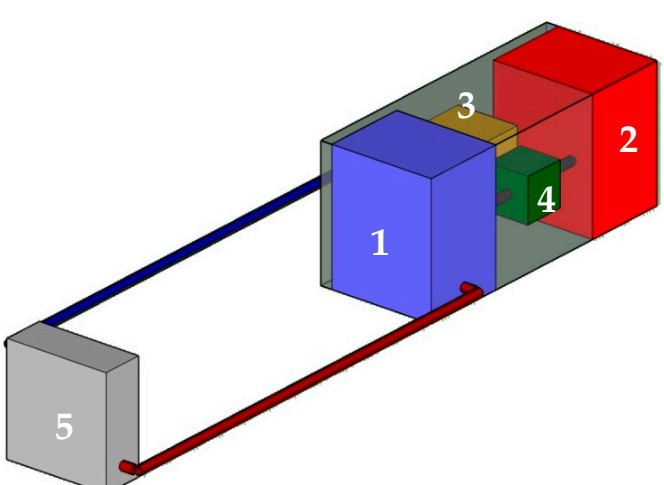

**Figure 11.** The layout of the ORC plant with the radiator. 1. Condenser; 2. Evaporator; 3. Expander; 4. Pump; 5. Vehicle radiator.

### 7. Conclusions and Future Works

An ORC plant for a small engine has been designed. The working fluid R245fa has been considered. Water is not competitive due to the small amount of mass flow rate, which does not permit the use of commercial devices for these applications. Another problem with water, as a working fluid, is that the required power by the pump is high. The working fluid R245fa allows the use of a small condenser and a lower average general pressure. Commercial devices can be used for the pump and the expander, whilst for the condenser, a brand-new prototype must be proposed. In the beginning, it was assumed to be possible to use the cooling system of the vehicle itself; this hypothesis was later abandoned because the water temperature was too high. Furthermore, the ORC and the Diesel engine would be, in this case, interlinked, creating potential trouble for the engine.

A strong limiting factor for the ORC plant, which affects its performance, is that the cooling water temperature is fixed by environmental conditions. A precautionary approach established that 40° represents the typical summer condition. The consequence is that the working fluid needs a higher condensing temperature, which leads to higher working pressure. The general efficiency of the plant is 7%, which is a typical range for small rate ORC systems.

The power input in a combustion engine is shared between the power provided to the auxiliary system, the cooling system, the exhaust, and finally the wheels. Because of the settings of this plant when used in a vehicle, a reduction in fuel consumption, though at a percentage that can now just be theoretically estimated, is nevertheless predictable. The purpose of proposed ORC plant is to provide work to the auxiliary system, for example, the lights system, leading to the higher efficiency of the ICE, thanks to the fact that more power is yielded to the wheels. The power output, in a general ICE engine, is roughly 30%; with the ORC unit installed, the efficiency of the plant should increase to 35%.

**Author Contributions:** Conceptualization, R.C.; methodology, R.C. and G.P.; software, R.C.; validation, G.P.; formal analysis, R.C. and G.P.; investigation, R.C. and G.P.; resources, R.C.; data curation, G.P.; writing—original draft preparation, R.C.; writing—review and editing, R.C. and G.P.; visualization, G.P.; supervision, R.C. and G.P. All authors have read and agreed to the published version of the manuscript.

**Funding:** No external funding.

**Institutional Review Board Statement:** Not applicable.

**Informed Consent Statement:** Not applicable.

**Data Availability Statement:** Not applicable.

**Conflicts of Interest:** The authors declare no conflict of interest.

### Nomenclature

| Symbol, Units | Description |
| --- | --- |
| CAD | Computer-aided design |
| CFD | Computational fluid dynamics |
| EV | Electric vehicle |
| EUDC | European urban drive cycle |
| FCEV | Fuel cell electric vehicle |
| HEV | Hybrid electric vehicle |
| KERS | Kinematic energy recovery sys |
| LPG | Liquefied petroleum gas |
| $\dot{m}$ (kg/s) | Mass flow rate |
| P (kW) | Power |
| T (K) | Temperature |
| W | Work |

| **Greek symbol** | |
| --- | --- |
| ε | effectiveness |
| ρ | density |

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
