# Peer review of "Condenser Design for On-Board ORC Recovery System"

_applsci, doi:10.3390/app11146356_

Round 1
Reviewer 1 Report
1- In the abstract, the purpose of study is not clear: not determined clearly if the ORC is going to study or the condenser itself.
2- English is very bad and complex sentences make it difficult to understand.
3- Nomenclature misses Description part!
4- Sentence 49 is just repeated!
5- A bunch of terms and concepts are explained without referring to where they are utilized in the model and how. These explanations can be provided as an appendix
6- What is the utilized software OR the coding platform?
7- Is it modeling or simulation?
8- Where is the model description? An algorithm of the model is, at least, required!
9- Lines 112 - 115: how are these values determined? Are they experimental data or theoretical values OR just assumptions?
10- Figures 3 and 6 are repeated. They can be referred to the Figure 2.
11- Line 160: what is the tank?
12- lines 162-164: are not understandable
13- Lines 192-194: the thickness of tubes is not considered? Whay?
14- Lines 227-231: How the numbers are obtained? These data should be the results of optimization. What is the procedure?
15- Equation 15 lacks in term "l" as it is explained in the followed text.
16- Line 351: Why did you change the Nu equation?
17- Lines 389-392: What are the provided intervals and why did you introduce them? Where are they utilized?
18- Line 408: one of the ρ parameters is missing in the explanation.
19- Line 452: which term OR terms go under iteration?
20- In figure 12: determine the components of the plant with some arrows.
21- Lines 612-613: How about the evaporator? is it available commercially?

Author Response
Please see the attachment.
Regards, Giuseppe Piras

Reviewer 2 Report
This manuscript presents a condenser design for an organic Rankine cycle system in vehicle application. The literature review can be improved. The condenser model is presented in detail and is the strength of this manuscript. However, the design procedure should be highlighted to avoid confusion. Overall, the manuscript is well written and here are my comments:
- The main claimed research gap is the lack of 5kW-level ORC system in literature. To my knowledge, the ORC WHR in vehicle application has been booming since 2010 and the power production level is in the range of 1kW – 15kW. The authors should at least give some examples of 100kW-level ORC system and existing 5kW-level ORC system.
- The focus of this study is condenser design. The authors should conduct literature review on condenser design in introduction section. Then you should clearly highlight the difference (and pros/ cons) between this study and existing literature in condenser design. Literature review is important to avoid the unnecessary repetition of work and takes advantage of others’ work. You need to prove this work is meaningful enough, which will then intrigue readers.
- Even though the equations are explained well, it is hard to follow how the numerous iterations are started and finished. The big picture and design procedure should be given in detail. What is the criteria for the design? How many numerical iterations are used?
- There is a space at the beginning of the second bullet point about the Eq. (16)
Author Response

(The authors gave the same response as above.)

Round 2
Reviewer 1 Report
Line 119: Do you mean that you have an experimental setup and data is driven from it? OR, you had just guessed some theoretical data?
If data are derived experimentally, you should clearly mention them.
English is very poor. Many conversions of nouns and adjectives are required.
Author Response

(The authors gave the same response as above.)

Round 3
Reviewer 1 Report
English should be improved (not satisfying yet)
Introduction is too short
More results may support the research further
Author Response
Please see the attachment.
Regards
